# Significance of the Surgical Treatment with Lymph Node Dissection for Neuroendocrine Tumors of Thymus

**DOI:** 10.3390/cancers15082370

**Published:** 2023-04-19

**Authors:** Naoko Ose, Soichiro Funaki, Takashi Kanou, Toru Kimura, Eriko Fukui, Eiichi Morii, Yasushi Shintani

**Affiliations:** 1Department of General Thoracic Surgery, Osaka University Graduate School of Medicine, Suita-shi 565-0871, Osaka, Japan; 2Department of Pathology, Osaka University Graduate School of Medicine, Suita-shi 565-0871, Osaka, Japan

**Keywords:** neuroendocrine tumors of thymus, surgical treatment, lymph node dissection

## Abstract

**Simple Summary:**

Neuroendocrine tumors of the thymus (NETTs) are rare and have a poor prognosis. Complete resection would be the best method to achieve a radical cure, although a standardized treatment method has not been established. Additionally, no standard criteria have been developed to determine the necessity of tumor resection alone or total thymectomy, and the same is true for lymph node dissection. We reviewed 40 patients who underwent surgical resection, including lymph node dissection and biopsy. The overall survival (OS) rate was better in the resected cases but with no significant differences in histological grade and pathological stage. NETTs frequently develop lymph node metastasis; thus, lymph node dissection seems necessary to achieve complete resection, especially for local control. Recurrence frequently occurs, but complete resection and multidisciplinary treatment would be effective because of the relatively good OS.

**Abstract:**

Background: Neuroendocrine tumors of a thymus (NETTs) are rare, accounting for approximately 2–5% of all thymic epithelial tumors, and have a poor prognosis due to frequent lymph nodes or distant metastasis. Methods: We retrospectively reviewed the clinicopathological background and outcomes of 40 patients who underwent resection or surgical biopsy with histologically diagnosed NETTs from 1986 to 2022. Results: The most common pathological type was atypical carcinoid. Surgical resection was performed in 35 patients, with lymph node dissection in 33 and surgical biopsy in five. The overall survival (OS) rate for all patients was 81.4% and 52.3% at 5 and 10 years, respectively. The 2-year survival rate was 20% for the biopsy group, which was significantly worse than that of the resected group (*p* < 0.001). The relapse-free survival rates were 61.7% and 37.6% at 5 and 10 years, respectively, in 34 patients with complete resection. The univariate analysis revealed better the OS rate in the resected cases but with no significant differences between histological grade, lymph node metastasis, tumor size, or Ki67 index. Conclusions: Surgical complete resection is considered to improve prognosis regardless of histologic type. NETTs frequently develop lymph node metastasis, thus, lymph node dissection seems necessary for complete resection.

## 1. Introduction

Neuroendocrine tumors (NETs), which originate from neuroendocrine cells, develop in organs throughout the body due to the distribution of these cells. The most common organs are the pancreas and gastrointestinal tract, found in about 60% of NETs, followed by the lungs and bronchi in 30% [1]. Prevalent organs are reported to vary from country to country. NETs also occur in the thymus, but the frequency is rare, with an incidence rate of only 0.4% [2,3]. The NETs of the thymus (NETTs) account for approximately 2–5% of thymic epithelial tumors [4]. NETTs are classified as low, intermediate, and high-grade tumors, with the low-grade tumors being referred to as typical carcinoids (TC), intermediate-grade tumors being atypical carcinoids (ATC), and high-grade tumors being large cell neuroendocrine carcinoma (LCNEC) according to the 2015 World Health Organization (WHO) classification. According to the 2021 WHO classification statement, NETTs are classified as an independent category, not one of the thymic carcinoma types, but the grade and histological definition and classification remain the same as the 2015 classification [5]. ATC is the most common type of NETT [2]. Its prognosis is poorer than those of gastrointestinal and pulmonary NETs and is said to be similar to that of thymic carcinoma [6]. Tumor size [7] and histopathologic subtype [8] are reported as prognostic factors. Although lymph nodes and distant metastases appear with high probability, the efficacy of chemotherapy or radiotherapy is reported to be insufficient for NETTs, and complete resection is important [3,9,10,11,12,13,14,15,16]. The prognosis has become poor in the progressive stage, but this can be explained by the confounding effect of the possibility of complete resection [10]. Thus, combined resection of another organ invasion or lymph node dissection is valuable [16]. The effect of lymph node dissection on prognosis improvement remains unclear [16,17]. Video-assisted thoracic surgery (VATS) and robotic-associated thoracic surgery (RATS) have been reported to be useful regarding surgical approaches for thymic malignant epithelial tumors [18], but no reports have examined NETTs alone. Whether induction or adjuvant therapy should be performed also remains controversial [10,19]. A previous report demonstrated the efficacy of adjuvant therapy [20], whereas many studies described a limited effect [3,6,7]. Currently, there are no established treatment recommendations for NETTs. Moreover, the number of cases reported in the literature is small, with very few reported cases from a large database [3,10,21]. Thus, the present study investigated the treatment outcome and prognosis of 40 cases of NETTs with a central pathology.

## 2. Materials and Methods

Forty patients who had undergone surgical resection or biopsy at Osaka University Hospital and ten associated centers between January 1986 and August 2022, who were histopathologically diagnosed with NETTs, and whose progress were available in the medical records, were included in this study. After accumulating the data about the tumor and the location of lymph node or distant metastases, the staging was performed according to the TNM classification based on the 8th edition of the UICC stage classification [22]. Individual informed consent was waived because of the retrospective study design. The study protocol was approved by the Ethical Review Board for Clinical Studies at Osaka University (control numbers: 10026-3 and 13469).

### 2.1. Histopathological Diagnosis

All cases were diagnosed by a pathologist (E.M.) according to the WHO classification of tumors of the thymus [4,5]. The primary lesions were used in 39 cases, and the resected lesion at the time of recurrence was found in one case. All tumor tissues were fixed in 10% formalin and embedded in paraffin, and 4-μm thick sections were routinely stained with hematoxylin and eosin. For immunohistochemical analysis, 4-μm thick serial sections were stained with CD56, synaptophysin, and chromogranin A to evaluate neuroendocrine differentiation and with Ki-67 to evaluate cell proliferation.

### 2.2. Surgical Treatment

All contents related to the indication of surgery and surgical treatment, including the resection area, the extent of lymph node dissection, and approaches, were left to the surgeon and the institution. The lymph node groups were classified using the lymph node map reported by Detterbeck et al. [23]. The patients in whom the thymectomy and intra- and peri-thymic lymph nodes were removed were considered to have undergone group 1 dissection (ND1), and the decision to perform group 2 dissection of the deep mediastinum (ND2) was left to the institution’s policy. The specific dissected lymph node station has not been investigated.

### 2.3. Statistical Analysis

Data are expressed as the mean ± standard deviation. The comparisons between the two groups were made using a Mann–Whitney U-test, with a chi-square test used for categorical variables. The survival rate was calculated using the Kaplan–Meier method from the date of surgery to the time of death for any causes (overall survival [OS]), recurrence (relapse-free survival [RFS]), and the last visit (censored OS and RFS). The log-rank test was used to assess the differences between subgroups. The probability values of <0.05 were considered significant. The hazard ratio and confidence limits were estimated for each variable using the Cox univariate model. All analyses were performed using the JMP15.0.1 statistical software package (SAS Institute Inc., Cary, NC, USA).

## 3. Results

### 3.1. Patients’ Characteristics and Surgical Treatment

Table 1 shows the patients’ characteristics and treatment. Fifteen patients (37.5%) were symptomatic, with chest pain and cough being the most common symptoms. Three patients had endocrine neoplasia 1 (MEN1), and only two patients (5%) presented with tumor-associated symptoms, such as ectopic adenocorticotropic hormone (ACTH) and Cushing’s syndromes. Among the studied population, nine cases were TC, 17 were ATCs, three were LCNECs, and 11 were small cell carcinomas of the thymus (SCTCs). 18F-Fluorodeoxyglucose positron emission tomography (FDG-PET) was performed in 17 patients both for differential diagnosis and detecting metastasis, and the median standardized uptake value (SUVmax) was 5.1. Induction therapy was performed on three patients who were each diagnosed with LCNEC, SCTC, and thymic carcinoma.

Thirty-five and five patients underwent surgical resection and biopsy only, respectively. Thirty-four of the 35 patients achieved complete resection, and one patient underwent resection of the primary tumor with a single bone metastasis. Thymectomy was performed in 33 and tumor resection in two. Lymph node dissection was performed in 33 cases. Complicated resections were performed in 15 cases, in whom the organs resected included the lung, pericardium, phrenic nerve, superior vena cava, brachiocephalic vein, and parathyroid gland, with none of the cases undergoing arterial resection. The surgical approach used was VATS in four cases and median sternotomy, including a hemi-clamshell incision in the remaining cases. Partial thymectomy was performed in only two cases, and total thymectomy was performed in the remaining cases. Sixteen patients underwent ND2.

### 3.2. TNM Factors

Table 2 shows the clinical and pathological stages according to the TNM classification [19]. There was no significant difference in the stage among the patients stratified according to the histological type. Eleven patients had lymph node metastasis. Five cases with a preoperative diagnosis of N0 developed pathologically lymph node metastasis, including N2 metastasis in three cases. There were three cases with multiple lymph node metastasis. Nine cases became advanced pathological stage, with lymph node metastasis being the most common cause in five patients.

### 3.3. Outcome

Postoperative adjuvant therapy was performed in 13 cases. Eleven patients received radiation therapy (RT) to the mediastinum alone, one patient received chemoradiotherapy (CRT), and one patient received somatostatin analogs. Regarding the recurrence site, patients who received adjuvant therapy had either distant metastasis or dissemination, and there was no recurrence in the irradiated area. The median follow-up duration was 62 (1–342) months. Four of the five biopsy-only cases died of primary disease within 1 year after the examination. Among the 34 patients who underwent complete resection, 15 patients had recurrence, with distant metastasis being the most common, followed by bone and lung metastases and then dissemination (Table 1). Lymph node metastasis occurred in only one case in the mediastinum cephalic to the dissected area. Post-relapse therapies included RT, chemotherapy, somatostatin analogs, peptide receptor radionuclide therapy (PRRT), and molecular-targeted agents. The median survival after relapse was 53 (43–110) months.

### 3.4. Analysis of Survival 

The OS rate for all patients was 81.4% and 52.3% at 5 and 10 years, respectively (Figure 1a). The 2-year survival rate was 20% for the biopsy group, which was significantly worse than that of the resected group (*p* < 0.001) (Figure 1b), but no significant difference in the 2-year survival rate among the groups stratified by histological type (Figure 1c). There was also no significant difference between the pathological TNM stage. In 34 patients with complete resection, the RFS rates were 61.7% and 37.6% at 5 and 10 years, respectively (Figure 1d), which worsened with increasing the p-TNM stage (*p* = 0.0332) (Figure 1e). The RFS rate was not significantly different among the groups stratified by histological type (Figure 1f).

In the univariate analysis, the OS rate was better in the resected cases (*p* = 0.0002), with stage 4 cases having a poorer prognosis than stage 1 and 2 cases. There were no significant differences among histological grade, lymph node metastasis, tumor size, or Ki67 index (Table 3).

## 4. Discussion

NETs can develop in any organ, but the frequency of occurrence, histological subtype, functional tumor or not, and prognosis vary depending on the primary organ. NETTs rarely occur, but thymic carcinoid arising from MEN1 is common. It has been reported that the incidence of MEN1 was only approximately 8% [21], and solitary occurrences are more common. Epidemiologically, NETTs are more common in men, with an average age of 40–50 years [3,7,24,25]. Thymic carcinoid with MEN1 is more common in male smokers [26,27]. In this study, the fact that 74% of men and only 20% of women were smokers suggests that being a male smoker may be a risk factor for NETTs.

Since patients of NETTs are often asymptomatic, imaging examination will be important validating tools for treatment. It is difficult to distinguish thymoma or thymic carcinoma from NETTs by imaging alone. FDG-PET is useful in differentiating thymomas from thymic carcinomas [28], and NETTs have a high accumulation of FDG [29]. Hyperaccumulation of tumors should be suspected in patients with thymic carcinoma or NETTs, and a biopsy should be performed. In pancreatic and gastrointestinal NETs, FDG-PET is used in combination with somatostatin receptor scintigraphy (Octreoscan) to diagnose the grade of NETs [30] because a carcinoid tends to have low FDG accumulation [31], but it is increased in neuroendocrine carcinomas with high proliferative potential. Octreoscan is used for NETTs in the evaluation of the indications for PRRT for thymic carcinoids [32], but there are no reports of its use in combination with FDG-PET for diagnosis. As ATCs may also show a high SUVmax value of 16, NETTs may show hyperaccumulation even in carcinoids. Thus, it is considered that FDG-PET is useful not only for differentiating thymoma from carcinoma but also for detecting distant and lymph node metastases. 

Since immunostaining is useful for the diagnosis of NETs, preoperative biopsies, such as CT-guided or chamberlain biopsies, should be performed in cases of large tumors, suspected lymph node metastasis, or invasion of other organs. The preoperative diagnosis of NETTs would be useful when considering the treatment strategy, such as the chemotherapy regimen, surgical approach, and need for lymph node dissection. However, it should be noted that pathological diagnosis can be difficult in some cases. We have previously reported a case in which the diagnosis of NETT was changed to squamous cell carcinoma or from a carcinoid to SCTC [15] and experienced different diagnoses like thymoma, not NETTs in preoperative biopsy or rapid intraoperative diagnosis. 

Many variables, such as tumor size, histological grade, paraneoplastic symptoms, Masaoka stage, TNM stage, surgical resection, and Ki67 index, were reported as prognostic factors [2,3,7,11]. In our series, there was no significant difference in the survival rate among groups stratified by tumor size or Ki67 index, and the patients who underwent complete resection showed a better prognosis. Complete resection is a prognostic factor in thymic carcinoma [33,34,35], and many studies on NETTs also reported that complete surgical resection is a prognostic factor [3,10,11,12,13,14,15,16]. In this study, there was no significant difference in the OS among the groups stratified by pathological stage. This may be due to the small number of cases enrolled in the study or due to the fact that cases with complete resection, even in those with stage 4 tumors with solitary lymph node metastasis near the tumor, were included. 

Whereas complete resection would be the best method to achieve a radical cure, there are no standard criteria to determine whether tumor resection alone or total thymectomy is necessary. 

Although some reports suggested that major vessel replacement [16,36] or lymph node dissection does not improve survival, lymph node dissection and resection of invaded organs are essential to achieve complete resection [16]. On the other hand, peripheral lymph node dissection was recommended [17], and N2 diagnosis can have more options, such as whether adjuvant treatment should be added after surgery or more careful follow-up is required because NETTs frequently have lymph node metastases [9,16,19]. In this study, there were upstage cases that were considered clinical-N0 but turned out to be pathological-N1 or N2. Only one case had lymph node recurrence; in this case, ND2 was performed, and p-N2 was diagnosed, but a single lymph node metastasis was found on the cephalic side of the dissected area 6 years after surgery. Lymph node metastasis was noted in stage 4b cases. There was no significant difference in the OS between the different groups with complete resection stratified by stage or lymph node metastasis; we speculate that a relatively good prognosis and local control around the tumor may be expected by complete resection with lymph node dissection in stage 4b cases due to lymph node metastasis. 

There is no clear consensus on induction therapy [6]. The usefulness of preoperative induction therapy has only been demonstrated in case reports [37]. In our hospital, cisplatin-based chemoradiotherapy is performed using a regimen similar to that for pulmonary NETs when LCNEC or SCTC is diagnosed. We have reported the resected case of LCNEC who achieved Ef3 due to induction chemoradiotherapy [37]. Induction therapy may be beneficial to patients who may be difficult to resect on the first operation, thereby increasing the probability of complete resection.

The benefit of RT as a postoperative adjuvant therapy is controversial. One study reported that it is effective [20], but several studies showed that it is not effective or that it resulted in several side effects [3,6,7,16]. The addition of postoperative RT could be useful as local control because no local recurrence, including that in the mediastinum lymph node, occurred in the RT cases, but the number of RT cases enrolled is too small to conclude that it is indeed effective; thus, a future study on this issue is warranted. The efficacy of postoperative chemotherapy is currently unknown.

Regarding the treatment after recurrence, the efficacy of standard systemic therapy has not been established due to the rarity of NETTs, so they will be treated as gastrointestinal and pulmonary NETs. In addition to the treatment with platinum-based combination chemotherapy with etoposide [11], somatostatin analogs, everolimus [38], and PRRT are effective for carcinoids. Since recurrence within 8 years after surgery is common [10] and the RFS rate consistently declines up to 10 years in this study, recurrence at >5 years after surgery should also be noted. Recurrence frequently occurs, but given the relatively good OS, multidisciplinary treatment would be effective because of the increasing treatment options available for NETs.

The major limitation of the present study is its retrospective design with a small number of cases. Therefore, it is undeniable that the information obtained is limited or insufficient, for example, stations of dissected lymph node stations. Although a prospective analysis is desirable, it is difficult to conduct, such as studies at a single institution owing to the rarity of NETTs; thus, a global study will be necessary. In addition, since only cases with surgical biopsy or resection were included in the study, the results are not sufficiently comparable to those of cases receiving actual medical treatments without surgical resection. A median sternotomy approach with lymph node dissection of aorticopulmonary and peritracheal lymph nodes is performed at our institution for NETTs and thymic carcinoma. Treatment strategies, including surgical techniques and area of lymph node dissection, are not standardized. The results of VATS and RATS cases should be reviewed to determine the appropriate approach. Future studies will be needed to analyze prognostic impact due to approach, amount of resection, RT, chemotherapy, and somatostatin analogs because we have not adequately verified the details both of surgical and non-surgical treatment. Given that the cases that were pathologically proven to be NETTs by re-diagnosis according to the 2015 WHO classification were enrolled, we believe that our study results are reliable as long-term treatment outcomes of pathologically confirmed NETTs.

## 5. Conclusions

Complete surgical resection is considered to improve the prognosis of tumors, regardless of their histologic type. NETTs often develop lymph node metastasis; thus, lymph node dissection seems necessary to achieve complete resection and accurate staging. In addition to total thymectomy with lymph nodes, dissection is recommended for these NETTs. Long-term follow-up would be necessary because recurrence can take place even after a long period after surgery.

## Figures and Tables

**Figure 1 cancers-15-02370-f001:**
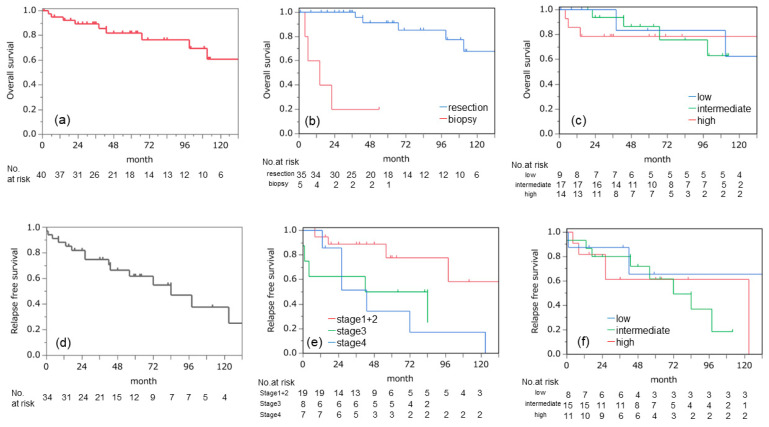
(**a**–**f**) Kaplan-Meier curves with patients according to (**a**) overall survival of all cases; (**b**) overall survival for resection or biopsy groups; (**c**) overall survival for histopathological type; (**d**) relapse-free survival of all cases; (**e**) relapse-free survival for pathological TNM stage; and (**f**) relapse-free survival for histopathological type.

**Table 1 cancers-15-02370-t001:** Patients’ characteristics, treatment, and outcome.

Patients Cheracteristics	Cases (*n* = 40)	Surgical Procudures	Cases (*n* = 40)
Sex (M/F)	35/5	Type of Surgery	
Age	53.7 ± 12.3 (17–80)	only biopsy	5
Symptom		resection	35
none	25 (62.5%)	R0	34
positive	15 (37.5%)	R2	1
Smoking history		lymph node dissection (*n* = 35)	
yes	26 (65.0%)	none	2
never	12 (30.0%)	ND1	17
unknown	2 (5.0%)	ND2	16
Functional tumor		Recurrence after resection ( *n* = 35 )	
yes	2 (5.0%)	no	18
no	38 (95.0%)	yes	17
MEN1	4 (10%)	regional	6
Outcome		diatant	8
alive	30	regional+diatant	3
dead of primary disease	8		
dead of another disease	2		

MEN1; Multiple endocrine neoplasia type 1, ND; lymph node dissection.

**Table 2 cancers-15-02370-t002:** Tumor characteristics.

	Cases (*n* = 40)
size (cm)	6.77 ± 4.0 (1.5–18)
Histological diagnosis	
typical carcinoid	9
atypical carcinoid	17
LCNEC	3
small cell carcinoma	11
Ki-67 labeling index (%)	25.5 ± 24.0 (1–90)
clinical WHO classification	
T1a/T1b/T2/T3/T4	16/0/10/11/3
N0/N1/N2	34/3/3
M0/M1a/M1b	37/3/3
stage 1/2/3a/3b/4a/4b	15/10/7/2/1/5
pathological WHO classification	
T1a/T1b/T2/T3/T4	18/6/1/14/1
N0/N1/N2	29/5/6
M0/M1a/M1b	35/2/3
stage 1/2/3a/3b/4a/4b	18/1/8/5/8

LCNEC: large cell neuroendocrine carcinoma.

**Table 3 cancers-15-02370-t003:** Univariate analysis of overall survival.

Variables	Hazard Ratio	95% Confidence Interval	*p* Value
Sex			
male	1.000		
female	3.811	0.90–16.1	0.069
Grade			
low	1.000		
intermediate	1.310	0.23–7.56	0.76
high	2.260	0.39–13.0	0.36
Resection status			
resection	1.000		
biopsy	26.90	4.75–152.7	0.0002
pathological TNM stage			
1 + 2	1.000		
3	1.850	0.26–13.4	0.54
4	5.080	1.01–25.3	0.048
Lymph node metastasis			
negative	1.000		
positive	3.155	0.89–11.7	0.075
Tumor size			
<7 cm	1.000		
≥7 cm	1.636	0.40–6.63	0.49
upstage			
none	1.000		
yes	1.768	0.48–6.40	0.39
Ki67 index			
<10%	1.000		
≥10%	2.040	0.57–7.37	0.28

## Data Availability

The data presented in this study are available in this article.

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
