# Peer review of "Significance of the Surgical Treatment with Lymph Node Dissection for Neuroendocrine Tumors of Thymus"

_cancers, 2023, doi:10.3390/cancers15082370_

Round 1
Reviewer 1 Report
Ose et al have worked on establishing the efficacy and significance of surgical resection with biopsy and lymph node dissection for tumor prognosis of NETTs. The authors reviewed 40 patients that were that were diagnosed with NETTS based on the histology reports. The authors work displayed that the overall survival rate was better in cases with resection irrespective of factors such as tumor size and histological grade. The Kaplan-Meier curves were a nice visualization of the patient data analysis.
There are a few opportunities to improve the clarity and impact of the manuscript.
Page 2 Line 55 – Is imaging one of the prognosis tools for such tumors?
Page 2 Line 60 – “area” seems to be a typo in this sentence.
Page 2 Line 81 – The “4” at the end of sentence is a typo.
Page 3 Line 107 – The FDG-PET was for imaging/prognosis? Please add a sentence here to clarify.
Page 4 Line 132 – Please mention the full form of CRT.
Page 4 Line 132 – The authors looked into broad categories of treatment such as RT and somatostatin analogs but did they try looking into the specifics such as using which RT was used and maybe there was a difference in prognosis or OS based on the kind of specific RT used. Similarly if the different somatostatin analogs had different trends in terms of survival then that might add more impact to the analysis by the authors. Also nowadays radioligand therapy (RLT) is used for somatostatin receptors such as Lutathera. Did the authors cover any patients treated with RLT?
Page 6 Line 181 – FDG-PET instead of PDG-PET.
Author Response
Response to Reviewer 1
We thank the reviewer 1 for the kind comments and constructive criticisms which we used to improve the quality of our manuscript. We have responded to all the additional issues raised by this reviewer. Changes made were shown in red below and in the manuscript.
Response to Reviewer 1 Comments:
- Page 2 Line 55 – Is imaging one of the prognosis tools for such tumors?
Authors response:
Thank you for your question. Since NETTs are often nonfunctional tumors, most patients are asymptomatic and the tumor can be detected by imaging examination, even at recurrence. We think an imaging provides a chance to start treatment early, but it is difficult to say for prognostic contribusions. We added the comment in Discussion, page6 Line 196-197.
2) Page 2 Line 60 – “area” seems to be a typo in this sentence.
Authors response:
Thank you for your suggestion. We deleted the word.
3) Page 2 Line 81 – The “4” at the end of sentence is a typo.
Authors response:
Thank you for your suggestion. We deleted the word.
4) Page 3 Line 107 – The FDG-PET was for imaging/prognosis? Please add a sentence here to clarify.
Authors response:
Thank you for your question. The FDG-PET was done both for differential diagnosis and detecting metastasis. We added the description in Page3 Line 127.
5) Page 4 Line 132 – Please mention the full form of CRT.
Authors response:
Thank you for your suggestion. We added the full form of CRT in Page 5 Line155.
6) Page 4 Line 132 – The authors looked into broad categories of treatment such as RT and somatostatin analogs but did they try looking into the specifics such as using which RT was used and maybe there was a difference in prognosis or OS based on the kind of specific RT used. Similarly if the different somatostatin analogs had different trends in terms of survival then that might add more impact to the analysis by the authors. Also nowadays radioligand therapy (RLT) is used for somatostatin receptors such as Lutathera. Did the authors cover any patients treated with RLT?
Authors response:
Thank you for your comment. Unfortunately, we have not been able to collect information of the type of RT and somatostatin analogs. Only one case has been treated with PRRT. PRRT was listed as a post-recurrence treatment in 3.3 Outcome.
As your comment is very valuable, we added the sentence about the details of treatment as a future issue in the limitation Page8 Line2890292.
7) Page 6 Line 181 – FDG-PET instead of PDG-PET.
Authors response:
Thank you for your suggestion. We modified.
Reviewer 2 Report
Authors reported the results of resection of neuroendocrine thymic tumors. Neuroendocrine thymic tumors are rare and the authors analyzed 40 patients. However, there are a number of concerns regarding the manuscript.
Title.
1- The title implies that, authors analyzed the impact of thymic tumor resection along with lymph node dissection. However, the article does not seem to focus on lymph node dissection. Authors did not compare the patients who underwent lymph node dissection with the one who had not lymph node dissection.
Abstract.
2- Authors did not give -even a very short- definition of inclusion criteria for surgery. There is no clue regarding the surgery and postoperative period of patients.
3- As conclusion, authors indicated that, ‘lymph node dissection is necessary’. However, there is no comparison comparing the patients who underwent lymph node dissection or not.
Introduction.
4- Authors did not give any evidence regarding the resectability of thymic neuroendocrine tumors. Which stages should be resected? Also, even the fact that lymph node dissection was given as the investigated factor, publications about lymph node positivity in neuroendocrine tumors were not mentioned in the Introduction section.
5- Surgical indications and surgical approaches such as open (i.e. via sternotomy), robotic and VATS procedures as well as neoadjuvant therapy for the stage III tumors were not summarized in the Introduction section.
Materials and Methods
6- Authors did not indicate the inclusion and exclusion criteria for surgical resection. Also, they did not indicate the important surgical challenges regarding the resection. Which stages were deemed as operable. Since there were metastatic patients reported in Table 2, authors seem to accept no boundaries for resection. However, it is plausible to predict that, many patients might undergo incomplete resections.
7- Authors did not give the surgical approaches that were utilized in the study. There are a number of patients who seem to undergo vats surgery, but the criteria for this option were not given.
8- The performed lymph node dissection methodology was not reported in the Materials and Methods section. Did authors remove paratracheal lymph nodes (i.e.N2 lymph nodes) in all patients? Were aorticopulmonary and anterior mediastinal (periaortic) lymph nodes dissected?
9- If authors did not accept any contraindication for resection, there must have been some patients with ‘open-close’ surgery (exploratory) due to aortic and/or main pulmonary arterial invasion intrapericardially.
10- Did authors perform pleurectomy/extrapleural pneumonectomy for the patients with pleural metastases?(i.e. M1 patients?)
11- Which type of thymic resections were performed in the patients? Tumor resection? thymothymectomy? extended thymectomy?
Results
12- Authors presented that, patients who underwent resectional surgery had better survival. However, patients who had resectional surgery seem to have lower stage tumors (i.e.stage II and III) compared to those undergoing biopsy only. I think, there is a confounder factor such as stage defining the survival of the resected patients.
13- Authors did not document the resected/dissected lymph node stations.
14- There is no survival analysis based on lymph node dissection. Authors should have performed a univariate and multivariable analyses in order to define the prognostic role of lymph node dissection.
15- The characteristics of lymph node positivity were not given in the manuscript. Were there patients with multiple N2 disease or multiple N1 disease?
16- I think, typical carcinoid tumors of the thymus should be evaluated separately due to their relatively less indolent nature. Probably, most operable patients had typical carcinoid tumors.
Discussion
17- In discussion section, authors did not seem to focus on the resectability of the tumors, lymph node dissection methodology and/or surgical approaches.
18- Authors did not seem to conclude any practical conclusion regarding the resectability, surgical approach or lymph node dissection. For instance, what would be the best protocol for the patients with neuroendocrine thymic tumors? Which patients can be operated via robotic assisted or video-assisted surgery? Authors did not give ‘take-home’ messages that can be extracted from the study.
Author Response
Response to Reviewer 2-
We thank the reviewer 2 for the many kind comments and constructive criticisms which we used to improve the quality of our manuscript. We have responded to all the additional issues raised by this reviewer. Changes made were shown in red below and in the manuscript.
Response to Reviewer 2 Comments :
1)Title.
1- The title implies that, authors analyzed the impact of thymic tumor resection along with lymph node dissection. However, the article does not seem to focus on lymph node dissection. Authors did not compare the patients who underwent lymph node dissection with the one who had not lymph node dissection.
Authors response:
Thank you for your comment. We use this title because complete resection with lymph node dissection was performed in most of the resectable cases. The title was changed at little.
2)Abstract.
2- Authors did not give -even a very short- definition of inclusion criteria for surgery. There is no clue regarding the surgery and postoperative period of patients.
Authors response:
Thank you for your suggestion. We added the description about surgery and lymph node dissection in Page 1 Line 27-28,30.
3) 3- As conclusion, authors indicated that, ‘lymph node dissection is necessary’. However, there is no comparison comparing the patients who underwent lymph node dissection or not.
Authors response:
Thank you for your comment. We consider lymph node dissection is necessary ‘for complete resection’. But comparison between with or without lymph node dissection could not be possible because only two patients without lymph node dissection.
4) Introduction.
4- Authors did not give any evidence regarding the resectability of thymic neuroendocrine tumors. Which stages should be resected? Also, even the fact that lymph node dissection was given as the investigated factor, publications about lymph node positivity in neuroendocrine tumors were not mentioned in the Introduction section.
Authors response:
Thank you for your important suggestion. We added the sentence with reference about resectability and lymph node dissection in Page 2 Line 65-68.
5) 5- Surgical indications and surgical approaches such as open (i.e. via sternotomy), robotic and VATS procedures as well as neoadjuvant therapy for the stage III tumors were not summarized in the Introduction section.
Authors response:
Thank you for your important suggestion. There is no report for comparison of approach in only NETTs cases. We added the sentence about approach in Page 2 Line68-72.
6) Materials and Methods
6- Authors did not indicate the inclusion and exclusion criteria for surgical resection. Also, they did not indicate the important surgical challenges regarding the resection. Which stages were deemed as operable. Since there were metastatic patients reported in Table 2, authors seem to accept no boundaries for resection. However, it is plausible to predict that, many patients might undergo incomplete resections.
Authors response:
Thank you for the comment. Unfortunately, since this was a multicenter, retrospective study, the indication for surgery was left to each institution. We added the srtatement in 2.2. This study included one patient with a single bone metastasis, but the reason for the choice of resection was unknown.
7) 7- Authors did not give the surgical approaches that were utilized in the study. There are a number of patients who seem to undergo vats surgery, but the criteria for this option were not given.
Authors response:
Thank you for the comment. As above, the criteria for selecting the procedure and approach are different at each institution.
8) 8- The performed lymph node dissection methodology was not reported in the Materials and Methods section. Did authors remove paratracheal lymph nodes (i.e.N2 lymph nodes) in all patients? Were aorticopulmonary and anterior mediastinal (periaortic) lymph nodes dissected?
Authors response:
Thank you for the comment. We described the area of lymph node dissection in Page 3 Line 99-103.
But we have not been able to get information which specific lymph nodes were dissected. We added the comment in Page3 Line106-107.
9) 9- If authors did not accept any contraindication for resection, there must have been some patients with ‘open-close’ surgery (exploratory) due to aortic and/or main pulmonary arterial invasion intrapericardially.
Authors response:
There were no patients with aorta or main pulmonary artery, but superior vena cava resection was done in 1 patient. In our institution, the use of cardio-pulmonary bypass is allowed for complete resection for thymic cancer and NETTs. We have experience using it in thymic carcinoma, but not yet in NETTs. However, other facilities in this study have a different criteria of indication, there may be only inspection was done. The discussion of major vessel replacement was stated in Page 7 Line 233-235.
10) 10- Did authors perform pleurectomy/extrapleural pneumonectomy for the patients with pleural metastases?(i.e. M1 patients?)
Authors response:
There were no patients with dissemination. Only biopsy was done in 2 M1a patients.
11) 11- Which type of thymic resections were performed in the patients? Tumor resection? thymothymectomy? extended thymectomy?
Authors response:
Thank you for the comment. This was also left to the institution. Actually, thymectomy was performed in 33, tumor resection in 2 and no extended thymectomy. We added this to the results Page 3 Line132-133.
12)Results
12- Authors presented that, patients who underwent resectional surgery had better survival. However, patients who had resectional surgery seem to have lower stage tumors (i.e.stage II and III) compared to those undergoing biopsy only. I think, there is a confounder factor such as stage defining the survival of the resected patients.
Authors response:
As you mentioned, there was a bias because the biopsy cases could not resected due to the advanced stage. Essentially, it should be compared to more cases without resection. We added the comment as the limitations Page 8 Line282-285.
But Fillosso et. al. said this could be explained by the confounding effect of whether or not complete resection is possible. The comment was added about this in Page 2 Line 66-67.
13) 13- Authors did not document the resected/dissected lymph node stations.
Authors response:
Thank you for the comment. We have got information which specific lymph nodes were dissected because some cases were operated on decades ago, and there was no information available.
We added the comment in limitation part, Page8 Line 279-280.
14) 14- There is no survival analysis based on lymph node dissection. Authors should have performed a univariate and multivariable analyses in order to define the prognostic role of lymph node dissection.
Authors response:
Thank you for your suggestion. Survival analysis based on ND was not possible because only two patient without anterior mediastinal lymph node dissection.
15) 15- The characteristics of lymph node positivity were not given in the manuscript. Were there patients with multiple N2 disease or multiple N1 disease?
Authors response:
Pathological N stage was listed in Table. There were 3 cases with multiple metastasis. But there was no detail data about lymph node station. We added this description in Page3 Line 149.
16) 16- I think, typical carcinoid tumors of the thymus should be evaluated separately due to their relatively less indolent nature. Probably, most operable patients had typical carcinoid tumors.
Authors response:
Our co-author (Dr. Morii) says that a diagnosis may be changed to atypical if the cross section for examination is different, even if pathologically diagnosed as typical carcinoid. Since it has been reported that most NETTs are atypical carcinoid and pathological diagnosis did not affect the survival, it cannot be said that typical carcinoid is more common in operable cases.
17) Discussion
17- In discussion section, authors did not seem to focus on the resectability of the tumors, lymph node dissection methodology and/or surgical approaches.
18) 18- Authors did not seem to conclude any practical conclusion regarding the resectability, surgical approach or lymph node dissection. For instance, what would be the best protocol for the patients with neuroendocrine thymic tumors? Which patients can be operated via robotic assisted or video-assisted surgery? Authors did not give ‘take-home’ messages that can be extracted from the study.
Authors response:
Thank you for your important suggestion. Our conclusion is total thymectomy with lumph node dissection is necessary to achive complete resection. As you pointed out, we fully understand that the number of cases and information is insufficient (Page8 Line 278-280).
We consider the resection is feasible if complete resection can be achieved with great vessel replacement and lymph node dissection with added reference in Page 7 Line 233-235.
We could not examine the result of resection between the approach due to the small number of cases, so it is considered an issue for the future. We added as a limitation in Page 8 Line 273-287.
Reviewer 3 Report
The paper faces an interesting retrospective study on Neuroendocrine Tumors of the Thymus (NETTs), a particularly rare pathology, diagnosed and treated in a Reference Center (Osaka University Graduate School of Medicine) and 10 associated centers. It is a careful analysis of 40 patients who underwent surgical resection or biopsy over almost 36 years (January 1986 - August 2022). Given the rarity of NETTs (accounting for approximately 2 - 5 % of all thymic epithelial tumors) is a very interesting series.
The conclusions:
1) complete surgical resection is considered to improve prognosis of tumor;
2) lymph node dissection would be necessary to achive coplete resection and accurate staging ((especially in total thymectomy with ND1 and ND2;
are not particularly original.
However, the accurate analysis of a relatively large series justify the interest of the specialist in this paper.
Author Response
Response to Reviewer 3
We thank the reviewer for the kind comments and constructive criticisms which we used to improve the quality of our manuscript. We have responded to all the additional issues raised by this reviewer. Changes made were shown in red below and in the manuscript.
Response to Reviewer 3 Comments :
The conclusions:
1) complete surgical resection is considered to improve prognosis of tumor;
2) lymph node dissection would be necessary to achive complete resection and accurate staging ((especially in total thymectomy with ND1 and ND2;
are not particularly original.
However, the accurate analysis of a relatively large series justify the interest of the specialist in this paper.
Authors Response:
Authors response:
Thank you for your kind comment. As you mentioned, it is not original. However, since TNETs are rare tumors, I think it is important to accumulate data s even on a small scale. We added the comments as a limitation in Page8, Line278-292.
Reviewer 4 Report
Comparing 35 patients in the resection group vs only five patients in the pathology group is not a valid comparison. I do not agree with the assertive conclusion that the authors wrote. Instead, I would compare the outcomes in the resection group to whatever is in the literature and mention the difference in the management in this current cohort.
- "The 2-year survival rate was 20% for biopsy group, which was significantly better than that of the resected group (p < 0.001)" I think it was worse?
- I do not understand what this study adds to the literature that we do not know.
Author Response
Response to Reviewer 4
We thank the reviewer for the kind comments and constructive criticisms which we used to improve the quality of our manuscript. We have responded to all the additional issues raised by this reviewer. Changes made were shown in red below and in the manuscript.
Response to Reviewer 4 Comments :
Comparing 35 patients in the resection group vs only five patients in the pathology group is not a valid comparison. I do not agree with the assertive conclusion that the authors wrote. Instead, I would compare the outcomes in the resection group to whatever is in the literature and mention the difference in the management in this current cohort.
Authors response:
You are exactly right. We fully understand that we should make comparisons with cases without surgical treatment, which is why I have included it in the limination, Page7 Line278-292.
- "The 2-year survival rate was 20% for biopsy group, which was significantly better than that of the resected group (p < 0.001)" I think it was worse?
Authors response:
Thank you for your important suggestion. It was our mistake. We corrected.
- I do not understand what this study adds to the literature that we do not know.
Authors response:
Thank you for your important suggestion. We believe that this study is valuable because of the assurance of a pathological diagnosis, as there are many cases where the central diagnosis has changed the diagnosis according to 2015 WHO classification. However, since TNETs are rare tumors, I think it is important to accumulate data s even on a small scale.
Round 2
Reviewer 4 Report
I still do not agree with the conclusion: "lymph node dissection seems necessary for complete resection." Adding it to a limitation paragraph does not make it better. I think the story of the paper needs to be changed as I suggested before.